# Presepsin in Critical Illness: Current Knowledge and Future Perspectives

**DOI:** 10.3390/diagnostics14121311

**Published:** 2024-06-20

**Authors:** Paolo Formenti, Miriam Gotti, Francesca Palmieri, Stefano Pastori, Vincenzo Roccaforte, Alessandro Menozzi, Andrea Galimberti, Michele Umbrello, Giovanni Sabbatini, Angelo Pezzi

**Affiliations:** 1Department of Anesthesia and Intensive Care, ASST Nord Milano, Ospedale Bassini, 20097 Cinisello Balsamo, Italy; miriam.gotti@asst-nordmilano.it (M.G.); francesca.palmieri@asst-nordmilano.it (F.P.); andrea.galimberti@asst-nordmilano.it (A.G.); giovanni.sabbatini@asst-nordmilano.it (G.S.); angelo.pezzi@asst-nordmilano.it (A.P.); 2Department of Clinical Chemistry and Microbiological Analysis, ASST Nord Milano, Ospedale Bassini, 20097 Cinisello Balsamo, Italy; stefano.pastori@asst-nordmilano.it (S.P.); vincenzo.roccaforte@asst-nordmilano.it (V.R.); 3School of Medicine and Surgery, University of Milano-Bicocca, 20126 Milano, Italy; a.menozzi2@campus.unimib.it; 4Department of Intensive Care, ASST Ovest Milanese, New Hospital of Legnano, 20025 Legnano, Italy; michele.umbrello@fastwebnet.it

**Keywords:** presepsin, sepsis, critical care patients

## Abstract

The accurate identification of infections is critical for effective treatment in intensive care units (ICUs), yet current diagnostic methods face limitations in sensitivity and specificity, alongside cost and accessibility issues. Consequently, there is a pressing need for a marker that is economically feasible, rapid, and reliable. Presepsin (PSP), also known as soluble CD14 subtype (sCD14-ST), has emerged as a promising biomarker for early sepsis diagnosis. PSP, derived from soluble CD14, reflects the activation of monocytes/macrophages in response to bacterial infections. It has shown potential as a marker of cellular immune response activation against pathogens, with plasma concentrations increasing during bacterial infections and decreasing post-antibiotic treatment. Unlike traditional markers such as procalcitonin (PCT) and C-reactive protein (CRP), PSP specifically indicates monocyte/macrophage activation. Limited studies in critical illness have explored PSP’s role in sepsis, and its diagnostic accuracy varies with threshold values, impacting sensitivity and specificity. Recent meta-analyses suggest PSP’s diagnostic potential for sepsis, yet its standalone effectiveness in ICU infection management remains uncertain. This review provides a comprehensive overview of PSP’s utility in ICU settings, including its diagnostic accuracy, prognostic value, therapeutic implications, challenges, and future directions.

## 1. Introduction

The accurate identification of infection is crucial for effective treatment and control of infectious diseases, but its recognition is often challenging, as signs and symptoms overlap with other inflammatory disorders [1]. Current diagnostic approaches in intensive care units (ICUs) rely on microbiological culture, biochemical methods, and molecular techniques [2]. However, there is still not a universally accepted standard due to the significant limitations in sensitivity and specificity of these methods. Additionally, their implementation often necessitates expensive technologies and equipment, which may not be accessible to all laboratories [3]. In clinical practice, clinicians use various indicators to detect sepsis. Though “biomarkers” typically refer to laboratory tests aiding identification and treatment guidance, they extend beyond lab assays. Fever and leukocytosis serve as biomarkers, and the absence of fever has been linked to delayed sepsis recognition and poorer outcomes. Given the limitations of current biomarkers, there is an urgent need for a marker that is economically feasible, rapid, simple, reliable, specific, and sensitive for the diagnosis of infection [4,5,6,7]. Presepsin (PSP), also known as soluble CD14 subtype (sCD14-ST) [8], is a promising biomarker that has garnered interest for its potential role in the early diagnosis and management of sepsis in ICU [9]. CD14 is a member of the Toll-like receptor (TLR) family and holds substantial importance in recognizing ligands from both Gram-positive and Gram-negative bacteria, thus initiating the inflammatory response [10]. CD14 exists in two forms, one anchored to the membrane of monocytes/macrophages (mCD14), and the other soluble found in plasma (sCD14) [11]. Within plasma, sCD14 undergoes cleavage by cathepsin D, resulting in the formation of a smaller fragment referred to as PSP. Levels of PSP in plasma exhibit elevation following bacterial infections and decline post-antibiotic treatment [12]. Therefore, this molecule can be considered a marker of cellular immune response activation against pathogens. PSP secretion has also been associated with monocyte phagocytosis, suggesting that it could be measured in healthy, non-infected individuals [13]. Unlike traditional markers such as procalcitonin (PCT) and C-reactive protein (CRP), PSP specifically reflects the activation of monocytes/macrophages in response to infections [14].

Within the context of critical illness, limited studies have explored PSP in sepsis, revealing that the accuracy of PSP determination relies on the chosen threshold value. For instance, with a threshold set at 600 ng/mL, sensitivity reached 70.3%, and specificity was 81.3% [15]. However, when a higher threshold (>860 ng/mL) was used, sensitivity improved to 71.4%, albeit at the cost of reduced specificity to 63.8%. A recent meta-analysis highlighted the diagnostic potential of PSP for sepsis, indicating high sensitivity and specificity [16]. Nevertheless, it is not yet shown whether PSP alone can effectively be used as an infection marker in the ICU.

This review aims to provide a detailed overview of PSP’s utility in ICU settings, encompassing its diagnostic accuracy, prognostic value, therapeutic implications, challenges, and future directions. Principal investigation on presepsin in critical care setting are summarized in Table 1. A review of the literature was conducted to evaluate published articles documenting presespsin investigations in critical care setting. Six databases were searched: PubMed (1996–present), Embase (1974–present), Scopus (2004–present), SpringerLink (1950–present), Ovid Emcare (1995–present), and Google Scholar (2004–present). The search utilized keywords such as “ presepsin”, “presepsin in critical care”, “sepsis and presepsin”, “diagnostic presepsin”, and “prognostic presepsin” across these selected databases. Two authors (PF and PMU) retrieved full texts of relevant articles. All related titles and abstracts were reviewed, and full versions were obtained. Exclusion criteria included studies involving pediatric patients, policy statements, and guidelines. The quality of the retrieved articles was assessed through careful evaluation of their methodology, sample size, study design, and relevance to the topic of PSP (Figure 1).

## 2. Immunobiology of PSP

CD14 is a membrane glycoprotein encoded by chromosome 5q that was first described in 1990 [17]. It serves as the receptor for lipopolysaccharide (LPS)-LPS-binding protein complexes, primarily found on monocytes/macrophages and, to a lesser extent, on neutrophil leukocytes [18]. Activation of tyrosine kinases and mitogen-activated protein kinases leads to the transcription of inflammation genes and the release of cytokines [19]. Subsequent activation of the secondary inflammatory cascade and acquired immunity further stimulates macrophages, neutrophils, and endothelial cells to release numerous other cytokines and synthesize adhesion molecules [20]. This can lead to an intense systemic inflammatory response, with activation of coagulation and fibrinolysis mechanisms. The result of these defensive mechanisms can sometimes be disproportionate and counterproductive. As a results, serious syndromes may be developed such as systemic inflammatory response syndrome (SIRS), septic shock, disseminated intravascular coagulation, and multiorgan dysfunction [21].

## 3. Soluble PSP Form

In addition to the membrane-bound form of phagocytes, CD14 also exists in a soluble form, derived from secretion or cleavage by plasma proteases [22]. It plays a role in mediating the immune response to LPS in cells typically indicated as CD14-negative, such as endothelial and epithelial cells [23]. Other authors hypothesize that soluble forms of CD14 may modulate the innate response to bacterial endotoxins by transferring lipopolysaccharides from monocyte membranes to plasma lipoproteins [24]. Clinical studies on sCD14 show that the plasma concentration of these molecules significantly increases in septic patients, and this increase is related to the severity of the condition [25]. The mechanism of presepsin production are described in Figure 2.

The soluble CD14 subtype is the N-terminal fragment of sCD14 derived from the antibacterial phagocytic activity of monocytes-macrophages and has been identified as a reliable marker of ongoing infectious processes in sepsis [26]. The metabolism and excretion of PSP are influenced by renal function; therefore, special attention is needed in interpreting values in the presence of chronic renal failure [27]. More specifically, the concentration of PSP was elevated in patients undergoing hemodialysis, suggesting that a distinct threshold should be taken into account for these individuals. [28]. PSP concentrations are known to rise with age, necessitating careful consideration when assessing elderly patients [29]. Additionally, PSP values of newborns, children, and adolescents require particular attention [30]. Moreover, PSP levels can be influenced by the translocation of intestinal microbial flora [31,32]. Given that PSP is excreted through both the kidneys and the hepatobiliary system, elevated concentrations may be detectable [33]. These findings underscore the importance of establishing tailored threshold values for particular populations and circumstances.

## 4. Diagnostic Utility of PSP

PSP demonstrates rapid kinetics, with elevated levels detectable within a few hours of infection onset [34]. Its ability to distinguish between septic and non-septic systemic inflammatory conditions makes it a valuable tool for early sepsis detection, although limited research has explored its utility in critically ill patients. Moreover, the interpretation and cutoff of PSP should be carefully evaluated in critically ill patients, as they often simultaneously present multiple organ dysfunctions.

Godnic et al. [35] established a diagnostic threshold of 413 ng/L for identifying bacterial infections in ICU patients. They discovered that PSP exhibited a higher area under the curve (AUC) than PCT, albeit lower than CRP. In a separate study involving ICU patients, PSP displayed favorable accuracy in predicting sepsis, with sensitivity and specificity values of 84.6% and 62.5%, respectively [36]. Notably, these results correlated significantly with the APACHE II score. Sargentini et al. [32] demonstrated that though PSP effectively discriminates between septic and non-septic patients in the ICU, its performance is inferior to that of PCT. The ALBIOS trial [15] observed that individuals infected with Gram-negative bacterial infections exhibited higher PSP levels compared to those with Gram-positive infections. Additionally, patients with bacterial infections, as figured out by site or blood culture, demonstrated significantly elevated PSP concentrations compared to individuals with negative culture results or those for whom no culture data were available. Endo et al. [37] reported divergent findings, indicating no notable distinction between Gram-positive and Gram-negative bacterial infections and no significant variations in PSP levels between the blood culture-positive and culture-negative groups. Presently, there are few meta-analyses available assessing the diagnostic efficacy of PSP in comparison to these biomarkers. A recent study compared PSP with PCT for early sepsis diagnosis in critically ill patients and concluded that both markers have similar efficacy, suggesting their combined use [38]. The study enrolled more than a thousand patients with confirmed infection and critical illnesses such as acute respiratory distress syndrome and sepsis. The diagnostic accuracy for detecting infection was found to be comparable between PCT and PSP, with sensitivity values of 0.80 and 0.84 and specificity values of 0.75 and 0.73, respectively. Both biomarkers proved to be valuable for the early diagnosis of sepsis and the reduction of mortality in critically ill adults. Similarly, a multicenter prospective study revealed that PSP is more closely linked with SOFA and APACHE scores than PCT in the clinical assessment of patients in emergency departments and ICUs [39]. PSP was assessed as a prospective biomarker for bacterial infection decline among critical care patients [40]. In cases of clinical recurrence of sepsis, PSP levels remained elevated, whereas PCT levels normalized during the transient remission phase. A retrospective analysis of 100 patients with suspected infection admitted to the medical ICU compared different sepsis biomarkers’ efficacy in the diagnosis of sepsis as compared with clinical definition [41]. The sensitivity, specificity, and AUC for sepsis diagnosis were higher for PCT (>0.5 ng/mL): 87.1%, 53.3%, and 0.776, and >1 ng/mL, 70%, 70%) and PSP (77.1%, 66.7%, and 0.734), showing a similar efficacy in diagnosing sepsis. However, none of the three biomarkers studied were accurate in predicting ICU mortality. Similarly, Xiao et al. [42] compared the predictive ability of PSP and PCT for bacteremia in more than 500 patients. Interestingly, the AUC achieved using PCT levels (0.856) was significantly higher than that achieved using PSP (0.786), and the combined analysis of two biomarkers led to a significantly higher AUC for predicting blood culture positivity and gram-negative bacteremia.

The presence of persistently high PSP levels may serve as an indicator for clinicians to consider continuation of antibiotic therapy in patients with sepsis. Overall, despite these conflicting results, we can summarize that the efficiency of PSP depends on the considered cutoff value. The differences in reported cutoff values across various studies may stem from the heterogeneity of clinical environments, the sepsis criteria used (pre- or post-Sepsis-3), the design of the studies (prospective versus retrospective), the presence of comorbidities, or the type of sample used (plasma, whole blood, or serum) for measuring PSP.

## 5. Prognostic Value of PSP

The possibility of having early prognostic information in patients with suspected sepsis admitted to ICU could provide fundamental data that meet the clinical need for management and therapeutic differentiation based on risk stratification of major events and therefore prognosis.

A few studies showed how patients with sepsis and septic shock who present high levels of PSP upon admission have a significantly higher probability of death at 30 days [43]. Hence, circulating PSP concentrations at admission may be used to stratify the risk of mortality. For instance, PSP concentrations on day 2 and day 7 post-admission were found to be independently correlated not only with ICU mortality but also with short-term (28 days) and long-term (90 days) post-admission mortality [15]. Interestingly, the SOFA score was the only clinical variable associated with mortality in a multivariate analysis model that included procalcitonin. When procalcitonin was replaced with PSP in the model, the SOFA score was no longer significantly associated with mortality. The potential prognostic use of PSP determination appears to be a promising tool, albeit poorly evaluated in ICU. PSP was included in a population previously investigated [44] to assess the effectiveness of combing procalcitonin with a clinical score, the Multidimensional Prognostic Index (MPI), in stratifying the risk of one-month mortality [45]. Upon admission, both PCT and PSP were measured. The results indicated that MPI effectively stratifies mortality risk upon admission, but the median values of PCT and PSP did not. Consequently, these biomarkers did not demonstrate significant prognostic efficacy. However, only PSP adds prognostic value when measured in association with MPI in stratifying intermediate-risk patients compared to low-risk patients.

The data from this study are consistent with recent literature in a critical setting. The prognostic value of PSP has been confirmed in stratifying short-term mortality risk in patients with pneumonia [46]. In this report, in more than a hundred ICU patients, the authors showed that PSP and PCT were significantly higher in septic than in non-septic patients. Moreover, in half of the patients, PSP capability to diagnose pneumoniae was significantly better than PCT.

Jovanovic et al. [47] investigated the prognostic significance of PSP for ventilator-associated pneumonia (VAP) and sepsis in critically injured patients necessitating mechanical ventilation. Their research revealed that PSP levels were notably elevated in patients who developed VAP. Additionally, PSP levels were considerably elevated in patients with sepsis compared to those with either VAP or SIRS. Zhao et al. [48] discovered that PSP serves as an independent predictor of in-hospital mortality in a cohort of ARDS patients.

Similar results have been described in other categories of critically ill patients, such as patients undergoing cardiac surgery, patients with cirrhosis, neonates with suspected sepsis, and patients with acute renal failure. In a separate analysis from the ALBIOS trial [49], which enrolled patients with severe sepsis or septic shock in ICUs, PSP levels were independently associated with both the number and severity of organ dysfunctions or failures, as well as with coagulation disorders and ICU mortality. Of note is the observation that not only admission biomarker values but also changes in concentration during monitoring seemed to provide interesting prognostic information that differentiates PSP from PCT. Indeed, not only were PSP concentrations significantly higher in patients who die, but in this group, they remained consistently elevated during monitoring. This type of combined biomarker application provides additional rather than mutually exclusive information. In fact, PCT and PSP determinations should be placed both in patients with suspected sepsis and in patients diagnosed with sepsis/septic shock.

Then, Xiao et al. [50] assessed the prognostic impact of PSP in sepsis. Their findings indicated that using PSP to guide antibiotic therapy did not adversely affect 28-day and 90-day survival rates. This approach appeared to outperform other conventional infection-related biomarkers like PCT. Enguix-Armada et al. [51] explored the prognostic capabilities of CRP, PCT, mid-regional pro-adrenomedullin, and PSP, all measured within the first 24 h of ICU admission. They assessed 28-day mortality and ICU length of stay as outcome variables. However, the study did not find any prognostic significance for PSP levels measured within the initial 24 h.

In the study conducted by Brodska et al. [52], which included 30 consecutive patients admitted for sepsis to a mixed medical-surgical ICU, different results were obtained. The study concluded that PSP did not outperform traditional biomarkers like PCT, CRP, and lactate in predicting mortality among critically ill patients with sepsis and SIRS. Koh et al. [53] assessed the efficacy of PSP as a biomarker for predicting in-hospital mortality in 153 patients with sepsis or septic shock admitted to the ICU. Though PSP values were elevated in the non-survivor group compared to the survivor group, the ROC analysis revealed poor performance of PSP in prognosticating sepsis outcomes. PSP levels exceeding 1176 pg/mL exhibited a sensitivity of 66.7% and specificity of 61.1% in predicting in-hospital mortality. More recently, Zhou et al. [54] investigated the predictive value of changes in PSP, PCT, CRP, and IL-6 levels for mortality. The study included 119 participants, with a mortality rate of 18.5%. In univariable Cox proportional-hazards regression analysis, changes in biomarkers indicated an increased risk of mortality (PSEP (>211.49 pg/mL): HR 2.70 (95% CI 1.17–6.22)). The composite concordance index for alterations in all four biomarkers was the highest (0.83, 95% CI 0.76–0.91), indicating the optimal performance of this panel in predicting mortality. In decision curve analysis, compared to the APACHE II and SOFA scores, the combination of the four biomarkers had a larger net benefit.

In summary, several studies conducted in critical care settings comparing the diagnostic and prognostic efficacy of PSP with CRP and/or PCT have produced solid results. PSP demonstrates a performance that generally aligns with that of PCT and appears to be a valuable parameter associated with patient outcomes.

**Figure 2 diagnostics-14-01311-f002:**
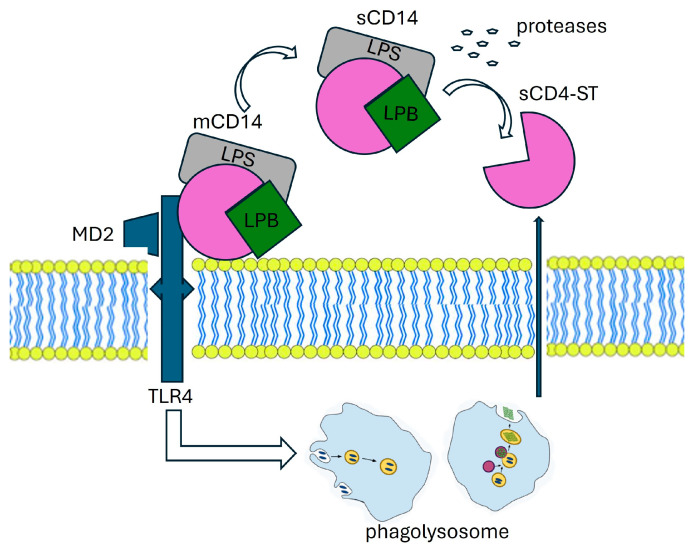
The mechanism of presepsin production.

The production mechanism of PSP involves various molecular players. CD14, found in two forms—membrane-bound (mCD14) and soluble (sCD14)—interacts with a complex, resulting in lipopolysaccharide (LPS) and lipoprotein binding protein (LBP). This complex, along with Toll-like receptor 4 (TLR4) and MD2, is internalized into a phagolysosome. Within this compartment, enzymatic processing facilitated by cathepsin D leads to the cleavage of CD14, resulting in the release of a small soluble peptide fragment known as soluble CD14 subtype (sCD14-ST), or PSP. This PSP fragment is subsequently released into the bloodstream via proteolysis and exocytosis. LBP: Lipoprotein Binding Protein, LPS: lipopolysaccharide, TLR4: Toll-like receptor, MD2:co-protein of TLR4.

**Table 1 diagnostics-14-01311-t001:** Principal investigation on presepsin in critical care setting.

	Study	Patients	Design	Main Findings
Diagnostic	Endo et al. [37]	207 ICU patients; suspected sepsis	Multicenter prospective study	PSP does not differ between patients with Gram-positive vs. Gram-negative bacterial infections. The sensitivity for discrimination of bacterial and nonbacterial infectious diseases of blood culture was 35.4% vs. PSP at 91.9%.
	Godnic et al. [35]	47 ICU patients	Comparative study three groups: SIRS, sepsis, septic shock	Bacterial infection showed statistical significance in PSP, CRP not in PCT. The severity of diagnosed SIRS was significantly associated only with PCT. Values of PCT were the only ones to predict SIRS severity and could distinguish between sepsis and severe sepsis or septic shock.
	Sargentini et al. [32]	21 ICU patients	Single-center, prospective observational study	ROC for the sepsis diagnosis was 0.945 PCT vs. 0.756 for PSP. Though PSP could effectively distinguish between septic and non-septic patients in the ICU, its performance was inferior compared to PCT.
	Xiao et al. [42]	522 septic patients	Retrospective multicentered	PSP vs. PCT in predicting bacteremia; the AUC achieved using PCT levels (0.856) was significantly higher than that achieved using PSP (0.786, *p* = 0.0200); combined analysis led to a significantly higher AUC.
	Kondo et al. [38]	3012 patients	Meta-analysis	No differences in both pooled sensitivities and specificities between PCT and PSP (0.80 vs. 0.84 and 0.75 vs. 0.73). Both biomarkers proved to be valuable for the early diagnosis of sepsis and the reduction of mortality in critically ill adults.
	Masson et al. [15]	100 ICU patients; severe sepsis or septic shock	Multicenter RTC	PSP, measured at day 1, was higher in non-survivors than in survivors. The evolution of PSP levels over time was significantly different in survivors compared to non-survivors; PSP concentrations on day 2 and day 7 post-admission were independently correlated with 28 days and 90 days post-admission mortality.
Prognsotic	Endo et al. [39]	103 ICU patients; sepsis or septic shock	Multicenter prospective study	PSP decreased on days 3 and 7 after ICU admission in survivors. PSP was more closely associated with SOFA and APACHE scores than PCT.
	Liu et al. [36]	859 hospitalized patients; SIRS	Single-center prospective observational study	PSP increased with sepsis severity. PSP demonstrated effectiveness in predicting sepsis (sensitivity and specificity 84.6% and 62.5%). PSP levels in septic patients were higher in non-survivors than in survivors at 28 days.
	Carpio et al. [43]	246 patients included	Single-center, prospective observational study. SIRS and/or sepsis vs. healthy	PSP levels were significantly different in patients with SIRS, sepsis, severe sepsis, and septic shock and showed strong association with 30-day mortality. Combination of PSP with MEDS score improved the performance for outcome prediction. PSP values in the course of the disease were statistically different between non-survivors and survivors.
	Klouche et al. [46]	144 ICU patients	Observational prospective study	PSP and PCT were significantly higher in septic than in non-septic patients. The prognostic value of PSP in stratifying short-term mortality risk in patients with pneumonia has been confirmed. In the patients admitted for acute respiratory failure, the accuracy of PSP to diagnose sCAP was significantly better than PCT.
	Zaho et al. [48]	225 ARDS patients	Multicenter prospective cohort trial sepsis-related ARDS vs. non-sepsis-related ARDS	PSP was found to be an independent predictor of in-hospital mortality in sepsis-related ARDS. Patients with sepsis-related ARDS had higher PSP levels than patients with non-sepsis-related ARDS. ROC PSP (0.81) was significantly greater than that of PCT (0.62). Among patients with sepsis-related ARDS, PSP levels were significantly higher in non-survivors than in survivors.
	Brodska et al. [52]	60 ICU patients	Single-center observational prospective	PSP did not correlate with SOFA on day 1. PSP did not demonstrate superior performance compared to traditional biomarkers such as PCT, CRP, and lactate in predicting mortality among critically ill patients with sepsis and SIRS.
	Zhou et al. [54]	119 ICU patients	Retrospective study	Predictive value of changes in PSP, PCT, CRP, and IL-6 to for mortality; △PSEP (=PSEP_3-PSEP_0) > 211.49 pg/mL (hazard ratio (HR) 2.70, 95% confidence interval (CI) 1.17–6.22).
	Koh et al. [53]	153 patient’s septic and septic shock	Retrospective cohort survival vs. non-survival	PSP values elevated in non-survivor vs. survivor group. PSP levels exceeding 1176 pg/mL exhibited a sensitivity of 66.7% and specificity of 61.1% in predicting in-hospital mortality.
	Juneja et al. [41]	100 ICU patients	Retrospective	The sensitivity, specificity, and AUC for sepsis diagnosis were PSP 77.1%, 66.7%, and 0.734; for ICU mortality, the sensitivity and specificity were PSP 61.5% and 27.3%; PCT and PSP had similar efficacy in diagnosing sepsis. However, none of the three biomarkers studied were accurate in predicting ICU mortality.
	Yu et al. [55]	109 patients	Monocentric observational prospective Survival vs. non survival	PSP levels in the survival group decreased persistently, whereas they rose gradually in the non-survival group.
Theranostic	Xiao et al. [33]	656 patients	Multicenter prospective cohort trial	PSP to guide antibiotic therapy did not adversely affect 28-day and 90-day survival rates. Patients in the PSP group also had significantly more days without antibiotics than those in the control group.
	Masson et al. [49]	997 patients; severe sepsis/septic shock	Multicenter randomized trial	PSP concentration at admission was associated with SOFA score. PSP levels tended to decrease in patients with negative blood cultures and in those with positive blood cultures and appropriate antibiotic therapy, whereas they were raised in patients with positive microbiology and inappropriate antibiotic therapy.
	Sargentini et al. [40]	64 ICU patients	Single-center prospective observational study	PSP levels remained elevated in recurrent septic patients, and PCT levels normalized during the transient remission phase. The presence of persistently high PSP levels may serve as an indicator for clinicians to consider continuation of antibiotic therapy in patients with sepsis.

## 6. Therapeutic Implications

Current guidelines recommend initiating antibiotic therapy within one hour of sepsis diagnosis [2]. However, the duration of antibiotic treatment often depends on the physician’s judgment and may vary based on treatment protocols [56], leading in some cases to prolonged antibiotic use. Extended administration of antibiotics not only results in significant costs but also increases the risk of complications, mortality, and prolonged hospitalization [57,58].

PSP-guided therapy has shown promise in optimizing antibiotic use and guiding early interventions in septic patients [55]. However, this potential role has not been thoroughly investigated in the critical care setting. The ALBIOS sub-study suggested that PSP might offer valuable guidance for therapy in sepsis. Masson et al. [49] investigated the potential of PSP as a biomarker in sepsis, discovering that PSP levels tended to rise in patients with positive microbiology and inappropriate antibiotic therapy.

Growing evidence supports the potentially beneficial approaches of PSP-guided antibiotic escalation and de-escalation. Xiao et al. [50] conducted a multicenter prospective cohort study aiming to investigate the utility of PSP in guiding physicians in the decision to continue or discontinue antibiotics for septic patients. The primary objective was to assess whether a PSP-based strategy would be linked to a reduction in antibiotic duration among septic patients, measured by the number of antibiotic-free days within a 28-day period or the duration until the initiation of the first antibiotic course. The authors showed that among patients with sepsis, employing a PSP-based antibiotic prescription strategy was linked with notable reductions in antibiotic treatment duration, ICU or hospital length of stay, and hospitalization costs. Importantly, these reductions occurred without any increase in mortality, recurrent infection rates, or risk of worsening organ failure. This targeted approach to therapy has the potential to improve patient outcomes while minimizing antibiotic overuse and associated complications.

## 7. PSP and Non-Bacterial Infection

PSP, traditionally regarded as a biomarker for bacterial infections, is emerging as a significant molecule in non-bacterial infections as well. Recent studies are exploring the role of PSP in conditions such as viral and fungal infections. Some studies have shown an increase in PSP levels in patients with fungemia, with significant correlations to disease severity. In this study by Bamba et al. [59], the authors investigated the potential utility of PSP as a biomarker for fungal bloodstream infections, an area where its usefulness is less understood compared to bacterial infections. Plasma PSP levels were measured in patients with fungemia, and its association with disease severity was analyzed alongside C-reactive protein and procalcitonin. Results showed elevated PSP levels in patients with fungal bloodstream infections, with a significant correlation to disease severity as indicated by the Sequential Organ Failure Assessment score. Moreover, in vitro assays demonstrated that viable Candida albicans cells induced an increase in PSP levels, suggesting its potential as a biomarker for sepsis secondary to fungal infections. In an experimental investigation by Bazhenov et al. [60], PSP expression in peripheral blood mononuclear cells from 19 healthy volunteers was examined following challenges with Candida albicans lysate, lipopolysaccharide, or autologous serum. After 24 h, the supernatant PSP concentration was 59.8 pg/mL (range, 29.7–140 pg/mL) for lipopolysaccharide-treated cells, 84 pg/mL (range, 38.8–133 pg/mL) for Candida albicans lysate-treated cells, and 34.6 pg/mL (range, 18.5–81.8 pg/mL) for autologous serum-treated cells. This trend persisted after 48 h, with PSP concentrations in Candida albicans lysate-treated cells remaining 1.3-fold and 2.8-fold higher compared to lipopolysaccharide or autologous serum-treated cells, respectively.

Overall, the supernatant PSP concentration significantly favored Candida albicans lysate challenge over lipopolysaccharide (*p* < 0.05) or autologous serum (*p* < 0.001). Based on these preliminary intriguing findings, some authors [61] suggested an algorithm based on combined PCT and PSP testing. Elevated levels of both PCT and PSP may indicate bacterial sepsis, particularly Gram-negative bacterial sepsis or mixed infections. Conversely, non-diagnostic values of both biomarkers may safely exclude bacterial or fungal sepsis. Meanwhile, a disproportionate increase in PSP levels, coupled with normal or modestly elevated PCT concentrations, could suggest invasive fungal infection. This interesting approach needs further diagnostic tests to be considered to achieve a more precise etiological diagnosis.

## 8. PSP and COVID-19

In December 2019, a new zoonosis named COVID-19, caused by the new Severe Acute Respiratory Syndrome Coronavirus 2 (SARS-CoV-2), appeared in China. The disease, characterized by clinical manifestations similar to ARDS, provoked a terrible pandemia from 2020, with elevated numbers of critical patients that rapidly overcrowded the ICUs all over the world. Ot 17 March 2024, the WHO updated the number of deaths to 7,040,264 [62]. Zaninotto et al. [63] described a first case series of patients in whom PSP were dosed. The authors formulated these observations: (1) PSP levels were higher in patients who died; (2) PSP showed a statistically significant but poor correlation with CRP and PCT; (3) PSP levels were related to ICU LOS. From these first observations, PSP demonstrated a possible role in providing diagnostic and prognostic information in COVID-19 patients, even if the disease was caused by viral and not by bacterial pathogens. Generally, PSP values do not increase in patients with viral infections.

Some years before COVID-19 outspread, Ozlem Demirpence [64] described an increase in PSP in patients affected by Crimean-Congo hemorrhagic fever; the PSP levels were related to disease severity. The authors speculated that elevated PSP levels are likely associated with macrophage activation also in a non-bacterial disease. Some years later, after COVID-19 outspread, high PSP levels were detected even in the mild COVID-19 [65]. Yamazaky et al. [66] speculated that SARS-CoV-2 can directly infect monocytes to reduce CD14+/CD16− classical monocytes and increase CD14+/CD16+ intermediate monocytes, which have an increased phagocytic function, resulting in the release of cytokines including PSP in the early stage of the disease. On the other hand, the typical clinical presentation of COVID-19 was similar to ARDS, and Zhao and colleagues [50] described how PSP levels were considerably increased in patients with ARDS independently from the etiology, but patients with sepsis-related ARDS had notably higher plasma PSP levels than patients with non-sepsis-related ARDS. Following the first observation cited above, several studies investigated the role of PSP as a diagnostic and a prognostic tool in COVID-19 patients. Assal and colleagues [67] found that PSP levels were significantly elevated in patients presenting with severe COVID-19, and levels above 775 pg/mL were significantly associated with in-hospital mortality (sensitivity 73% and specificity 80%). They postulated that elevated PSP levels indicate poor outcomes and should alert the physicians in making decisions regarding intensive care monitoring and further interventions.

The same results were described by Kocyigit et al. [68], who found that PSP levels were significantly higher in patients with SARS-CoV-2. Moreover, there was a significant correlation between PSP and disease severity. Lippi and colleagues [69] published the results of a pooled analysis of six studies with a total of 420 COVID-19 patients, whom 173 (41.2%) with a critical form of disease. They found that PSP levels were increased by 2.74-fold in COVID-19 patients with severe illness compared to those without. Guarino et al. [70] published a meta-analysis of data from 707 patients from 15 studies, and they found that the pooled mean difference of PSP levels between high- and low-severity COVID-19 patients was 441.70 pg/mL (95%CI: 150.40–732.99 pg/mL). In another interesting study published by Dell’Aquila and colleagues [71], in a population of COVID-19 patients with acute respiratory failure in an emergency department, PSP was an accurate predictor of 30-day mortality. PSP achieved a sensitivity of 54% and a specificity of 92% for a cut-off value of 871 pg/mL. The AUC for the ROC curve was 0.85. The authors proposed that PSP’s high specificity could help in the early identification of patients who could benefit from more intensive care as soon as they enter the emergency department.

Fukui et al. [72] showed in a cohort of 201 patients the high prognostic value of PSP in non-severe COVID-19 patients, suggesting that PSP might be a highly sensitive indicator of immunological reactions against infectious antigens in the early stage of COVID-19 infection, and it might predict subsequent disease evolution. Different studies proposed a combination of clinical and biochemical markers of inflammation to better detect patients affected by COVID-19 with poor prognosis, underlining how high PSP levels in the first 7 days of hospital stay were a good biomarker of poor prognosis [73,74,75,76].

More accurately, Yamazaky et al. [66] evaluated PSP values at multiple time points as well as the change in values after admission for patients with COVID-19. The authors observed an elevation in PSP values in non-survivors over time; however, these elevations were not observed in survivors. Moreover, some non-survivor patients with COVID-19 showed renal dysfunction, so the authors adjusted PSP for renal failure, and significant differences in PSP values remained. These data indicated that PSP values might be used as predictive markers, apart from renal function.

In summary, PSP could be a useful tool in diagnosis and prognostication in COVID-19. The PSP levels and trend correlated with the severity and the evolution of the disease, so high PSP levels should alert the physicians in making decisions regarding intensive care monitoring and further interventions.

## 9. Conclusions

According to Sepsis-3, the new sepsis’s definition emphasizes the critical role of the host response to infection rather than simply focusing on the presence of the infection itself. Consequently, terms such as “bacteremia”, “fungemia”, and “bloodstream infection” are no longer synonymous with sepsis. This shift in perspective highlights the importance of biomarkers that capture the disproportionate “non-homeostatic” host response, rendering them more effective for the initial screening of septic patients compared to direct microbial identification from blood samples. The first step towards establishing PSP as a widely accepted biomarker would be its integration into laboratory parameters without excessive costs, similar to what has occurred with other markers in the past. This integration would facilitate routine assessment of PSP levels, enabling broader clinical use and further research into its diagnostic and prognostic utility in critical illness. Thus, further research is needed to further elucidate PSP’s clinical utility, refine its diagnostic and prognostic capabilities, and optimize its integration into sepsis management protocols in the ICU.

## Figures and Tables

**Figure 1 diagnostics-14-01311-f001:**
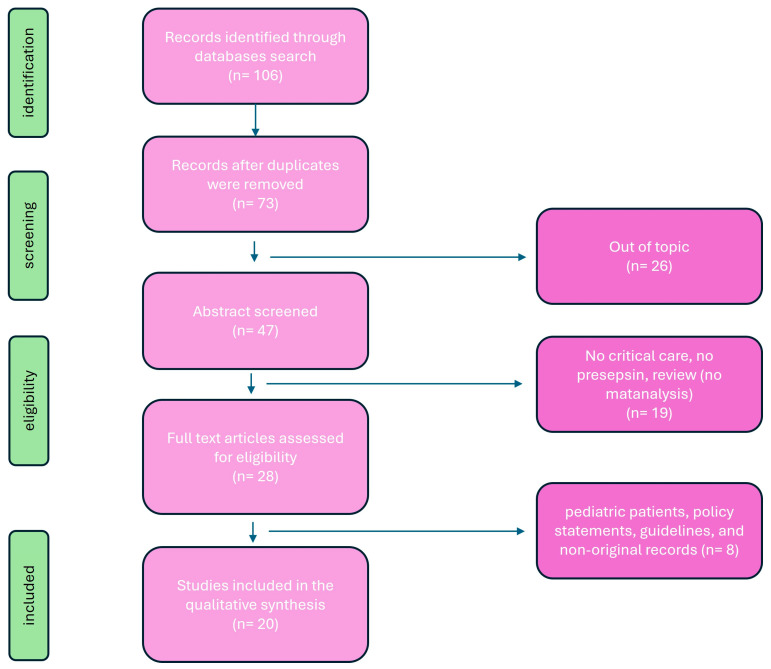
Flowchart of search strategy.

## Data Availability

No new data were created or analyzed in this study. Data sharing is not applicable to this article.

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
