# Peer review of "Presepsin in Critical Illness: Current Knowledge and Future Perspectives"

_diagnostics, 2024, doi:10.3390/diagnostics14121311_

Round 1
Reviewer 1 Report
Comments and Suggestions for Authors
This manuscript represents an overview of the current status of presepsin as a potential diagnostic and prognostic biomarker in the evaluation and management of individuals with critical illness, specifically sepsis and COVID-19. Overall, the authors have provided a comprehensive overview of the potential utility of presepsin in critical illness. However, some aspects of the manuscript should be addressed to improve the presentation of the data reviewed.
Specific Concerns and Suggestions
1. Overall, English grammar and syntax are very good throughout the manuscript. However, several sections are solid unroken prose for 1-2 pages of text. This format compromises the overall quality of the manuscript. The text should be revised into discrete, separate paragraphs to improve the quality of the manuscri\pt. I would also suggest that the authors consider adding subsections in eac of the surrents sections.
2. Table 1 should be divided into sections denoting investigation of presepsin as a 'diagnostic' biomarker and investigation of presespsin as a 'prognostic' biomarker, It would also be of interest if any studies have been performed that specifically address the utility of presepsin as a 'predictive' or 'theranostic' biomarker in critically ill patients.
3. During the initial part of the manuscript, the authors suggest/imply that presepsin elevation is specific for bacterial sepsis. However, at the end, the authors include a section on presepsin utility as a prognostic biomarker in COVID-19 , a viral cause of sepsis. Clearly, sepsis does not appear to be a specific for sepsis caused by bacterial infection. Has presepsin been evaluated for diagnostic/prognostic utility in other causes of virus-related sepsis, such as severe influenza, dengue, etc. How about fungal-infection related sepsis or protozoan-related sepsis (i.e., severe malaria)?
4. Suggest that the authors provide additional information/data to support the claim that elevated prespsis distinguishes infection-related sepsis vs. sepsis physiology not related to infection (formerly SIRS). The support cited is relatively weak for this clinically important distinction. Biologically, activation of monocytes/macrophages is an important feature of many diseases causing systemic inflammation and organ failure not caused by infection per se.
5. The manuscript would be improved by the authors providing more detail regarding proposed future studies that they view as important to definitively establish presepsin as an accepted biomarker in critical illness.
6. Line 60-62 appears to not be related to the subject of this manuscript: "A review of literature was conducted to evaluate published articles documenting perioperative ultrasound diaphragm evaluation ...."
Comments on the Quality of English Language
Overall, the quality of English language, both grammar and syntax, is quite good throughout the text. Current formatting according to standards of English writing is deficient and needs revision as noted above.
Reviewer 2 Report
Comments and Suggestions for Authors
Dear Authors,
I have read a review regarding the role of Presepsin in critical illness. While interesting, there are several points that need addressing:
1) This manuscript has a high similarity towards (https://www.intechopen.com/online-first/83909) and after manually checking each sentence, there is indeed a high similarity. Please fix this.
2) The keyword is weird, with numbers following each keyword.
3) Line 32 --> There is no gold standard. There is a gold standard, just that it is cumbersome with a high false negative. Please fix this.
4) What does DD stand for in line 62?
5) Please include a paragraph on what constitutes a good biomarker.
6) Please also include whether presepsin can be used to distinguish viral vs bacterial infection
7) A quick search using "Presepsin" and "ICU" on Pubmed already yielded 104 articles. Are the authors sure that using a combination of all the databases only yields 73 studies?
8) What is an RTC in Table 1?
9) The authors should also specify the inclusion criteria. Usually, a meta-analysis is not included in a primary result for a review as the meta-analysis includes the primary study included in this review anyway.
Comments on the Quality of English LanguageThere are numerous spelling and punctuation errors. For example, line 65 where the keyword is only "sepsi". This is just an example and please send the manuscript to a professional proofreader as other examples of errors are purposefully not listed here so I could compare whether this manuscript is actually sent to a professional proofreader or not.
Round 2
Reviewer 1 Report
Comments and Suggestions for Authors
The authors have revised the manuscript nicely to address my concerns/criticisms. The quality of the manuscript has been significantly improved. However, revised Table 1 was not included in the revised version that I downloaded. Also, the figures and table are now inserted into the body of text, rather than at the end of the manuscript. This should be corrected.
Author Response
We thank again the reviewer for his/her time and considerations.
We uploaded the new table.
The position of figures and tables will be adressed by the editorial team.
We will ask confirmation.
Reviewer 2 Report
Comments and Suggestions for Authors
Dear authors,
Thank you for addressing all the points. Good luck in your journey.
Comments on the Quality of English Language-
Author Response
We thank the reviewer for his time